# Autologous Stem-Cell Transplantation for High-Risk Neuroblastoma: Historical and Critical Review

**DOI:** 10.3390/cancers14112572

**Published:** 2022-05-24

**Authors:** Jaume Mora

**Affiliations:** Pediatric Cancer Center Barcelona, Hospital Sant Joan de Déu, 08950 Barcelona, Spain; jaume.mora@sjd.es; Tel: +34-934-192-800

**Keywords:** neuroblastoma, autologous stem-cell transplant, myeloablative therapy, hematopoietic stem cells, secondary neoplasm, survival outcomes, EFS and OS

## Abstract

**Simple Summary:**

The original idea that providing higher doses of cytotoxic agents will result in higher rates of tumor cell killing was proposed in the 1980s. Preclinical data supported clinical testing. Advancements in bone marrow and peripheral stem-cell support technologies during the 1980s and 1990s allowed for clinical developments that permitted testing the higher dose hypothesis in oncology patients. The results of almost 20 years of clinical trials proved the linear relationship between dosing and clinical outcome to be mostly inaccurate. As a consequence, the adult oncology field abandoned high-dose chemotherapy strategies by the turn of the 21st century. Neuroblastoma is the only pediatric extracranial solid tumor where high-dose chemotherapy has remained part of the standard management for high-risk cases. This systematic review aims to understand the historical reason for such an exception and analyzes data challenging the benefit of high-dose chemotherapy and autologous stem-cell transplants in the era of anti-GD2 immunotherapy.

**Abstract:**

Curing high-risk neuroblastoma (HR-NB) is a challenging endeavor, which involves the optimal application of several therapeutic modalities. Treatment intensity for cancer became highly appealing in the 1990s. Investigative trials assumed that tumor response correlated with the dosage or intensity of drug(s) administered, and that this response would translate into improved survival. It was postulated that, if myelotoxicity could be reversed by stem-cell rescue, cure might be possible by increasing the dose intensity of treatment. The principle supported autologous stem-cell transplant (ASCT) strategies. High-dose therapy transformed clinical practice, legislation, and public health policy, and it drove a two-decade period of entrepreneurial oncology. However, today, no ASCT strategies remain for any solid tumor indication in adults. As with most solid malignancies, higher dosing of cytotoxic agents has not resulted in a clear benefit in survival for HR-NB patients, whereas the long-term toxicity has been well defined. Fortunately, novel approaches such as anti-GD2 immunotherapy have demonstrated a significant survival benefit with a much less adverse impact on the patient’s wellbeing. On the basis of extensive experience, persisting with administering myeloablative chemotherapy as the standard to treat children with HR-NB is not consistent with the overall aim in pediatric oncology of curing with as little toxicity as possible.

## 1. Hematopoietic Stem-Cell Transplant: A Revolutionary Development from the 20th Century for Hematological Diseases. Historical Review

The era of hematopoietic stem-cell transplantation began after the first atomic bomb explosion, with the pioneering observations by Leon Jacobson et al. in 1949. Mice were protected from the lethal effects of ionizing radiation on bone marrow (BM) by protecting their spleens with lead [1]. Subsequently, Lorenz and colleagues showed that surviving lethal radiation could also be achieved by infusions of BM [2]. The discovery showing that all hematopoietic stem cells arise from pluripotent transplantable stem cells led to the use of BM transplantation as a treatment for hematological malignancies. In the years 1959–1965, three landmark papers were published that described the first experiences with transplantation of BM into leukemia patients [3,4,5]. In the first study, two patients with acute lymphoblastic leukemia received high-dose total body irradiation (TBI), followed by BM grafts from twin siblings [3]. This was the first example of patients given supralethal irradiation who showed hematological recovery.

The second half of the 1960s provided a significant evolution of high-dose myeloablative conditioning regimens. These consisted of maximally tolerated doses of TBI and/or chemotherapeutic agents such as cyclophosphamide, which not only served to kill cancer cells, but also suppressed the immune system of the host so that BM grafts would not be rejected [6]. The 1980s and 1990s showed how successful stem-cell transplantation could be safely applied with refined supportive care and broader application using alternative graft donors and sources. T-cell depletion and immunosuppressive agents were shown to minimize graft-versus-host disease. Drugs such as busulfan were identified as alternatives to TBI [7]. In 1990, E. Donnall Thomas won a Nobel Prize for his pioneering research on stem-cell transplantation for the treatment of hematopoietic diseases, laying the foundations of modern stem-cell replacement therapy for human diseases including cancer. In the year 2000 alone, approximately 18,000 patients worldwide were treated with myeloablative approaches, a strategy accepted as the standard of care for a range of specific indications.

## 2. The Basic Principle of Transplantation Expanded to Solid Tumors: More Is Not always Better

With the improvement of supportive therapy and sophistication in transplant-related techniques, treatment intensity for cancer became highly regarded in the 1990s partly because of the lessons from Hodgkin’s lymphoma. Higher cumulative dosage or higher dose intensity (DI) could be applied since myelotoxicity was no longer limiting in the face of stem-cell rescue. Investigative trials assumed that tumor response correlated positively with the dosage or intensity of drug(s) administered, and that this response would translate into improved overall survival (OS). Among the many preclinical studies, a steep log–linear dose–response curve was consistently documented for some cytotoxic agents (Figure 1) [8]. Increased chemotherapy drug dosage correlated with a log-fold increase in cell kill. It was postulated that, if myelotoxicity (the major dose-limiting toxicity of chemotherapeutic agents) could be reversed by stem-cell rescue, cure might be possible by increasing the dose or dose intensity of treatment. The principle supported autologous stem-cell transplant (ASCT) strategies. This linear–log relationship between chemotherapy agent dose and tumor cell killing suggested that, if the drug dose is increased without increasing toxicity to the patient, then a multiple log increase in tumor cell death would be observed [9,10]. Few prior clinical studies were available but suggested a positive relationship [9,10]. In addition to the limitation of extrapolating in vitro studies to human beings, there are underlying assumptions of this “cure by dosing” hypothesis, i.e., that the dose–response curve in patients is indeed linear, that drug resistance is overcome by increasing treatment intensity, that myelotoxicity is the only dose-limiting concern, and that response will translate into improved survival. Decades later, this hypothesis has been tested in multiple cancer types, and its success has been revealed to be disease-specific and qualified.

Chemoresistance, be it intrinsic or acquired, is a key factor impacting response and the ability to achieve a cure. In chemotherapy-sensitive solid cancers, a high initial response rate to chemotherapy is commonly followed by subsequent relapse and resistance to further treatment. Tumors fitting this model usually show a dose–response behavior to drugs given in the conventional dose range. In principle, the drugs causing this effect should be appropriate for further dose escalation, with dose-limiting toxicity (DLT) primarily affecting the hematopoietic organ rescuable with hematopoietic stem cells. Ideally, their nonhematologic effects should be limited, and morbidity and mortality should be minimal. Inherent to this approach is the assumption that resistance can be overcome with the increase in dose, which can be achieved with stem-cell support [11]. Alkylating agents are the most appropriate agents for dose escalation [10]. There was evidence from cell lines and animal models of a steep dose–response curve for these compounds [10], and resistance to alkylating agents had been overcome by a 5–10-fold increase in dose. Their DLT was primarily hematologic, and nonhematologic toxicities at high doses were non-overlapping. In contrast, anthracyclines are not suitable for dose escalation not only because of their cardiac and mucosal toxicities, but also because their cytotoxicity in preclinical studies leveled off. Antimetabolites and vinca alkaloids are also not suitable agents for high-dose therapy (HDT) not only because of their non-marrow toxicity limiting significant escalation, but also because their dose–response curves in vitro and in vivo are not linear [10]. Dose–response curves for most chemotherapy agents were determined using in vitro and in animal models, with tumor cell killing being directly proportional to both dose and duration of exposure (Figure 1, red line). Importantly, the steepness of the dose–response curve for most drugs showed a distinct plateau above which more drug does not appear to kill more tumor cells (Figure 1, green line) [7]. Furthermore, in humans, it is yet mostly unknown whether a plateau (Figure 1, green line) is reached when drugs are administered at the maximal tolerated dose (MTD).

The concept of HDT refers to pulsed chemotherapy, which achieves high peak drug concentrations that theoretically increase the overall exposure of tumor cells to supratherapeutic levels of cytotoxic drugs [12]. The concept of “dose intensity” (DI), however, is not the same as HDT. DI is defined as the amount of drug administered per unit of time (e.g., mg/m^2^/week). A dose-intensive regimen may or may not yield high peak concentrations (HDT). Continuous administration of cyclophosphamide might be quite dose-intensive, but it could be associated with lower peak concentrations and less acute toxicity than a similarly dose-intensive, pulsed, high-dose regimen. Different trials have shown the importance of dose intensity in the treatment of small round cell tumors including sarcomas [13,14] and neuroblastomas [15].

Myelosuppression associated with cytotoxic therapy often limited the amount of drug that could be administered. This limitation was overcome to some degree with the use of hematopoietic cytokines such as granulocyte colony-stimulating factor (G-CSF) and techniques to support hematopoiesis such as autologous peripheral blood stem cells and/or BM. Prospective studies in adults with solid tumors showed that dose escalation with cytokine support did not increase cure rates in patients with chemotherapy-responsive solid tumors (testis cancer, small-cell lung cancer (SCLC), advanced ovarian cancer, non-Hodgkin’s lymphoma, or bladder cancer [16]). The many reasons explaining this lack of benefit are as follows: (1) cytotoxic agents produced toxicity to nonhematopoietic tissues that limited the capacity to escalate doses further; (2) patients received extensive prior treatments that resulted in impairment of the BM function sufficient to blunt the response to cytokines; (3) the use of CSFs did not allow dose escalation beyond twofold higher than conventional doses.

Testicular germ cell tumors are among the most chemotherapy-sensitive solid tumors, and cisplatin appears to be the best single agent for this disease. If doubling the dose of cisplatin does not result in significant benefit, it would be unlikely that lesser increases in dosage above standard dose levels in other less chemotherapy-sensitive solid tumors would be beneficial. Indeed, preclinical studies suggested that 5–10× higher drug concentrations may be necessary to overcome drug resistance for most carcinomas [12]. By the end of 1990s, the adult oncology field concluded that HDT regimens with cytokine support were not to be considered standard therapy for solid tumors or lymphomas [16] despite the technical ability to deliver higher than standard doses of chemotherapy to patients with drug-sensitive cancers.

What about myeloablative therapies with BM support? Many clinical trials involving a variety of adult solid tumors were performed in the 1990s testing myeloablative therapy with BM transplant [11]. For the most common tumors (breast, ovary, and SCLC) compelling evidence to support the use of myeloablative chemotherapy in standard patient management did not emerge. Randomized data comparing standard therapy to HDT was found unconvincing. In breast cancer, eight randomized trials were reported, and the data did not support the use of HDT in breast cancer. In SCLC, insufficient evidence supported the change in practice. Selection bias was found to account for favorable results in uncontrolled studies in these tumors, as many authors pointed out by reviewing outcomes of HDT candidates who received only standard therapy [11]. HDT was ultimately determined to be ineffective and to have significantly greater toxicity. From its birth in the 1980s to its abandonment in the late 1990s, HDT transformed clinical practice, legislation, and public health policy, as well as drove a long period of entrepreneurial oncology. It also gave rise to one of the most serious cases of research misconduct of the 20th century [17]. Today, no ASCT strategies remain for any solid tumor indication in adults. For pediatric tumors, except for specific subgroups of lymphomas and germ-cell tumors, no other extracranial solid tumor indication of ASCT has been established as standard management for high-risk cases. ASCT strategies have failed to show benefit for Ewing sarcomas [18], Wilms tumors [19], non-rhabdomyosarcoma soft-tissue sarcomas [20], rhabdomyosarcomas [21], osteosarcomas [22], hepatoblastomas [23], and rhabdoid tumors [24].

As for neuroblastoma, similar controversies dismissed in adult cancer 25 years ago remain and are analyzed in the next section.

## 3. Why Only Neuroblastoma? The Original Mistake

In the 1980s and 1990s, studies in neuroblastoma suggested an improvement of outcome with myeloablative therapy in conjunction with autologous BM transplant [25,26,27,28,29,30,31,32,33,34,35,36,37,38,39]. It was the time when adult oncology was massively testing HDT strategies for solid tumors as previously reviewed. A common denominator of success in all these trials appeared to be the use of DI therapies; therefore, the identification of effective agents for DI was essential for optimal application of the principle. In order to select agents for DI in neuroblastoma, Cheung and Heller performed a retrospective meta-analysis looking at 44 trials between 1965 and 1989 including 1592 neuroblastoma patients [15]. The analysis concluded that dose intensity of etoposide and platinum had the greatest influence on response, EFS, and OS, more than doxorubicin or cyclophosphamide, while vincristine intensity had no effect. Twenty-one weeks of treatment duration was adequate, beyond which there was no clear gain. According to linear regression, the strongest *R^2^* for correlation with dose intensity was EFS, greater than that for objective response and OS. There was no indication that EFS or OS curves were asymptotic with increasing DI, raising a serious question of whether cure was within reach. The publication from 1991 led to the formal test of the role of DI in neuroblastoma in the form of sequential trials (reviewed in [39]) and, importantly, three randomized trials [40,41,42].

From January 1991 until April 1996, the CCG-3891 study enrolled patients with HR-NB in a randomized trial, becoming a landmark study published in 1999 [40]. A large, randomized study seemed the final test for the primary question of the CCG-3891 study, whether ABMT improved survival for children with HR-NB. The CCG-3891 study randomized patients in first CR after induction treatment to receive a conditioning regimen of carboplatin, etoposide, melphalan, and TBI, followed by an infusion of purged BM vs. continuation chemotherapy of three cycles of cisplatin, etoposide, and doxorubicin with ifosfamide [40]. The study concluded that the EFS 3 years after randomization was significantly improved among the 189 patients who were assigned to undergo transplantation compared to the 190 patients assigned to receive continuation chemotherapy (34% ± 4% vs. 22% ± 4%, *p* = 0.034). This initial report from 1999 prompted the worldwide adoption of myeloablative therapy (regardless of the conditioning regimen) with ASCT as the standard of care for all patients with HR-NB in first complete remission (CR).

Long-term results of the CCG-3891 study were reported 10 years later [43]. In the updated publication, the long-term follow-up of the randomized trial showed significantly better 5 year EFS and OS rates for myeloablative therapy with purged ABMT than for nonmyeloablative chemotherapy. Furthermore, the differentiation inducer 13-*cis*-retinoic acid (*cis*-RA) given after intensive therapy resulted in a significant improvement in 5 year OS rates, regardless of the type of consolidation. Surprisingly, all the statistical analyses initially published were corrected 5 years later [44]. The amended conclusion of the study was rephrased as “long-term follow-up of the randomized CCG-3891 trial showed significantly better 5 year EFS rate for myeloablative therapy with purged ABMT than for nonmyeloablative chemotherapy, although there was no statistically significant improvement in OS”. Neither myeloablative therapy with autologous hematopoietic cell rescue nor *cis*-RA given after consolidation therapy significantly improved OS. The amended results confirmed the predictions from the 1991 meta-analysis where maximal dose intensification of selective drugs over a short duration (HDT and ASCT) improved the outcome of patients with HR-NB with superior EFS but not OS [15].

Over three decades, three phase III randomized trials were completed [40,41,42] testing DI in HR-NB, and they all concluded modest improvements in EFS but no statistically significant improvement in OS in cases investigating (1) an increase in induction intensity, (2) single transplant for consolidation versus none, and (3) tandem transplant versus single transplant for consolidation [45]. A meta-analysis of ASCT and HR-NB published before the publication of the erratum in 2013 [46] and a subsequent update from 2015 [47] found no significant difference in OS between the treatment groups.

Another independent observation that reduced the likelihood of HDT to ablate NB that survived exposure during induction was the study from the COG documenting no advantage of ex vivo purging [48]. NB contamination of harvested stem cells was rare, suggesting that relapse could be attributed to the failure of myeloablative therapy to eradicate post-induction residual NB.

In conclusion, the accumulated data demonstrate that HDT and ASCT strategies improve EFS (delays relapse) but not OS (life expectation) of HR-NB patients. In light of these results, should ASCT remain as part of the standard management of patients with HR-NB? What does it mean for patients to undergo a highly toxic (and costly) treatment in order to increase EFS but not OS?

## 4. Event-Free Survival and Overall Survival

Overall survival (OS) is the gold standard for evaluation of treatment efficacy. It is the critical test for drug approval since it takes into account both safety and efficacy. In contrast, EFS is a surrogate endpoint and, thus, requires validation [49]. Unless improvement in quality of life can be certified, delaying asymptomatic events without improving survival is not clinically meaningful [49]. Does ASCT improve the quality of life of patients with HR-NB? If so, it should remain within the recommended backbone regimen until a new treatment able to increase OS becomes available. Nevertheless, a surrogate endpoint like EFS should be used with caution as the basis to establish standards of care.

After the CCG-3981 landmark study, there have been only two other reports on randomized ASCT trials for HR-NB [41,42]. Each showed an advantage of ASCT regarding EFS. The non-ASCT arms were clearly at a disadvantage, receiving no maintenance in one study [41] and only cyclophosphamide in the other [42]. The reported benefit in EFS in those studies also needs revision. For HR-NB, surveillance testing after completion of all treatment (including ASCT) was inconsistent and downplayed by most groups according to the view that relapse was incurable. In the post-ASCT cohort, relapse was usually discovered by symptoms from widespread disease and, thus, delayed in time. In contrast, non-transplanted patients would undergo frequent testing with the hope that retrieval with ASCT might still achieve cure. Therefore, relapses might be detected in asymptomatic patients, possibly more amenable to rescue treatments and, thus, earlier in time. This asymmetry of evaluation practices introduced biases in studies causing an artificially shorter EFS for the non-transplanted cohorts.

In the CCG-3981 landmark study and all randomized studies of HR-NB testing ASCT, OS failed to show significance. The only explanation of this discrepancy is that relapse after ASCT was invariably fatal. Taking into account the revised results of the landmark study and the critical importance of OS with very long follow-up, a reevaluation of ASCT for HR-NB should be undertaken. This is even more the case when considering that randomized ASCT studies carried out in 1982–1985 [41], 1991–1996 [40] and 1997–2002 [42] all fall short compared to current state-of-the-art therapy for HR-NB. They are based on the lower DI of the induction regimens, irregular use of radiotherapy, and lack of anti-GD2 immunotherapy, all being treatment modalities that have improved HR-NB disease control. The three randomized studies also preceded current salvage therapies and modern methodologies to detect resistant or recurrent NB (^123^I-metaiodobenzylguanidine scintigraphy and molecular quantitation of BM minimal residual disease). An important final consideration is that the term “transplant” was widely invoked in these studies (and protocols), but the variability of the cytoreductive agents used was extreme, ranging from a single agent (melphalan) to multiple agents plus TBI, which is no longer used for any solid tumor treatment including NB.

The high prevalence of late effects including second malignant neoplasms (SMN) following myeloablative therapy further undermines the use of ASCT [50]. It is important to remember that a 10% improvement in EFS means that 90% of patients are undergoing an unnecessary treatment with significant toxicities that will be life-long. A clear dose intensity relationship with SMN induction was established, and the mechanisms of therapy-related mutagenesis with expansion of premalignant clones were recently described in NB survivors [51]. Studies from the 1950s to 1980s including 544 patients with NB who survived at least 5 years showed a cumulative incidence of SMN of 2.2%, 3.6%, and 8.9% at 20, 25, and 30 years [52]. Studies from the 1970s to 1980s including 954 survivors of NB showed a cumulative incidence of SMN of 3.5% at 25 years and 7% at 30 years [53]. Studies from the 1980s to 1990s with 380 survivors of NB from MSKCC showed a 36 month cumulative incidence of AML of 7% [54]. Lastly, studies from the 1990s including 87 patients from three Chicago trials showed a 15 year cumulative incidence of SMN 34.2% [50], without evidence of plateau at 15 years.

Not only one but two high-dose chemotherapy and sequential transplants have been proposed as a consolidative strategy for HR-NB. In the sole reported randomized trial involving tandem transplant [45], the comparison was with single transplant (not with no transplant). Tandem showed better EFS, but OS was again not significantly different. The fundamental issue is whether transplant—single or double—is warranted for HR-NB according to the data available. A rethinking of the consolidation strategy for HR-NB would be consistent with the general sense among pediatric oncologists and recently among parents [55] that ASCT is unsatisfactory on the basis of both risk–benefit and cost–benefit analyses [56]. Few solid tumors, pediatric or adult, have been proven to benefit in terms of OS (not surrogate endpoints such as EFS). Neuroblastoma seems not to be an exception.

## 5. Anti-GD2 Immunotherapy and ASCT

Fortunately, chemo-resistant NB is highly responsive to treatments far less toxic than myeloablative therapy, namely, anti-GD2 antibody + granulocyte–macrophage colony-stimulating factor (GM-CSF) [57] ± low-dose chemotherapy [58]. It is noteworthy that two anti-GD2 mAbs—dinutuximab and naxitamab—have received approval from the Food and Drug Administration in the USA.

Immunotherapy targeting GD2 emerged with the aim to eliminate chemotherapy-refractory disease, similar to HDT strategies, in the 1990s and early 2000s. The murine monoclonal antibody (mAb) 3F8 underwent extensive preclinical testing and, when administered with GM-CSF, led to significant responses in patients with refractory NB [59]. Clinical trials of anti-GD2 mAb therapy demonstrated significantly improved EFS and, for the first time, OS when used after major responses to standard induction therapy and ASCT, leading to regulatory approval in the United States and Europe [60,61]. The problem is that the two strategies (ASCT and anti-GD2 immunotherapy) targeting minimal residual disease (MRD) have not been formally evaluated independently.

The only randomized controlled trial involved HR-NB patients who received either anti-GD2 mAb dinutuximab-containing immunotherapy or *cis*-RA after ASCT (ANBL0032) [60]. The primary objective was an intention-to-treat comparison. At 2 years, the estimated EFS was 66% in the dinutuximab-containing immunotherapy arm and 46% in the standard therapy group. A subsequent report provided longer follow-up data on survival [62]. The survival difference between the treatment groups remained statistically significant with a 5 year EFS of 56.6% for patients randomized to triple (dinutuximab, IL-2, and GM-CSF) immunotherapy versus 46.1% for those randomized to *cis*-RA only (*p* = 0.042). The 5 year OS was 73.2% versus 56.6% for triple immunotherapy versus *cis*-RA only patients, respectively (*p* = 0.045). The role of anti-GD2 immunotherapy after HDT and ASCT was clearly shown to be relevant for the survival of patients with HR-NB, and anti-GD2 immunotherapy was then adopted worldwide as the standard of care for all HR-NB patients in remission post ASCT. These results, however, question even further the role of HDT and ASCT in the era of anti-GD2 immunotherapy given that the landmark CCG-3891 study, as described earlier [40], did not show a benefit in OS when tested without anti-GD2 mAb. More recently, a detailed analysis by the SIOPEN of risk factors and the relevance for each of the sequential therapies within the HR-NB multimodal treatment program (chemotherapy, ASCT, and immunotherapy), revealed that the impact of immunotherapy on EFS is significantly influenced by stage and pattern of metastases, but only borderline related to ASCT [63].

Initial evidence of the superiority of anti-GD2 immunotherapy over HDT and ASCT for the management of patients with HR-NB came from the MSKCC group. This group had a great interest in and experience with transplant during the HDT era [28,34,35,36,37,38] but discontinued it in 2003 when their analyses of sequential studies with or without ASCT and/or mAb 3F8 showed no survival benefit [64]. In 2016, they first reported similar outcomes for HR-NB patients whose consolidative therapy of first CR/VGPR included 3F8/GM-CSF + isotretinoin with or without prior HDT and ASCT [65,66]. They were able to accrue a large cohort of more than 100 patients treated without ASCT, a remarkable group of patients given that ASCT has remained in all major studies since the year 2000 and continues today. Independently, in our institution, we were able to recruit a cohort of non-ASCT patients (*n* = 54) adding to that from MSKCC, increasing the population of HR-NB patients managed in the current era without ASCT and showing no difference in survival [67]. Recent advances in anti-GD2 immunotherapy, such as more potent mAbs [68], the lesser use of toxic IL-2 [61], and the systematic use of GM-CSF, may account for the current lack of survival advantage of ASCT. With all this evidence (or lack of it), discontinuing ASCT for HR-NB would be the logical step forward and consistent with the general consensus among (pediatric and adult) oncologists that this highly toxic treatment from the 20th century should no longer be recommended as long as anti-GD2 immunotherapy is broadly available to all children affected by HR-NB.

## 6. Conclusions

Curing high-risk neuroblastoma is a challenging endeavor which involves optimal application of several therapeutic modalities. Unfortunately, these may not be available in many centers and regions of the world. The first goal, as always in medicine, is primum non nocere (“first do no harm”); that is, if possible, avoid treatment that is worse than the disease itself. The balance between efficacy and toxicity should be carefully evaluated, especially in young children given their heightened susceptibility to long-term and all-too-frequent debilitating side-effects from chemotherapy and radiotherapy, which contrast strikingly with the absence of delayed toxicity from anti-GD2 mAbs. As with most solid malignancies, higher dosing of cytotoxic agents has not resulted in a clear benefit in survival of HR-NB patients, whereas the long-term toxicity has been extensively described. Fortunately, novel approaches such as those based on anti-GD2 immunotherapy have demonstrated a significant survival benefit with a much less adverse impact on a patient’s wellbeing. On the basis of extensive experience in recent years, most notably with mAbs, persisting with administering myeloablative chemotherapy as the standard to treat children with HR-NB is not consistent with the overall aim in pediatric oncology of achieving cure with as little toxicity as possible.

## Figures and Tables

**Figure 1 cancers-14-02572-f001:**
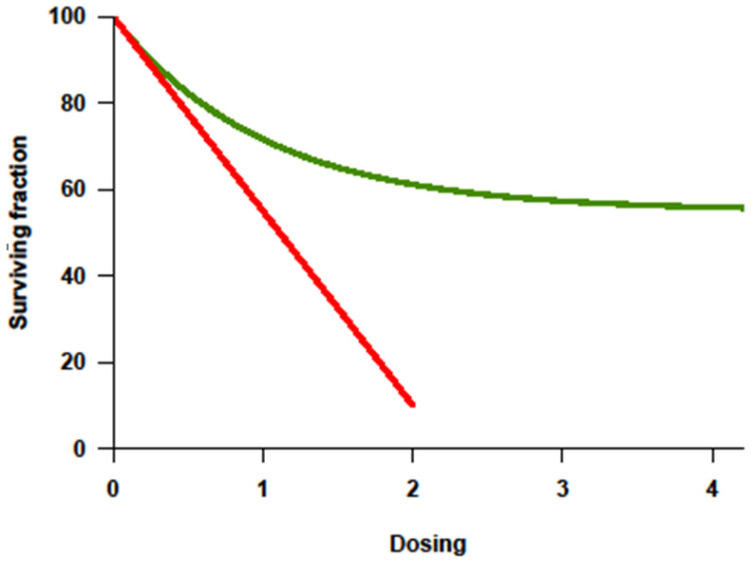
Dose–response curve showing log-fold increase in cell killing (*Y*-axis = survivor cells) with increase in dosage (*X*-axis) as the basis for high-dose therapy strategies. Adapted from original in Frei E III, Canellos GP: Dose: A critical factor in cancer chemotherapy. Am J Med 69:585–594, 1980. [9].

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
