# Peer review of "Autologous Stem-Cell Transplantation for High-Risk Neuroblastoma: Historical and Critical Review"

_cancers, 2022, doi:10.3390/cancers14112572_

Round 1
Reviewer 1 Report
Overall comments
This review describes the history and current state of autologous stem cell transplantation as neuroblastoma treatment. To my knowledge there are no other recent reviews of that type of treatment, which makes the review novel, interesting and useful. The author makes a strong case that in the era of immunotherapy high-dose chemotherapy followed by stem cell transplantation is no longer (if it ever was?) a valid treatment for neuroblastoma treatment. I feel convinced.
The author seems to be missing this review: Fish, J. D., & Grupp, S. A. (2007). Stem cell transplantation for neuroblastoma. Bone Marrow Transplantation, 41(2), 159–165.
Specific comments
The historical review is interesting and sets a nice background for the further part of the paper.
Line 84-85, please cite those few clinical trials.
Figure 1. Either the Figure has to be re-plotted or the wording has to be changed. The text says “increase in dosage/cell kill” but the figure shows a decrease in numbers (of surviving cells). Also, please explain what is red and what is green also in the legend.
Lines 140-142 – the statement that HDT is not efficient for solid tumors is crucial for the author so please cite the original research instead of just one review.
Line 147-148 citation please.
Again, lines 157-161, the author makes a strong case citing only one review paper, it’s necessary to add more citations, especially original publications.
Lines 255-263 – is this based on author’s clinical experience or some published data?
Lines 285-292 cite several studies of SMN in neuroblastoma patients, the results of which are very different. Why?
Author Response
Please see attached word document
Reviewer 2 Report
Dear Author
The topic of this review is very interesting for paediatric oncologists who treat children with neuroblastoma.
I have some questions and comments.
- What kind of the review has Author prepared?
- What methods were used to prepare this review?
- Simple summary - I don't agree with the statement "Neuroblastoma is the only solid tumor whereby high-dose chemotherapy has remained as a part of the standard management for high-risk cases". It should be changed.
- The paper is about ASCT for High Risk Neuroblastoma, which is a typical paediatric solid tumor. The section 2 "The basic principle of transplant expanded to solid tumors: more is not always better" applies to adults. It is necessary to include data about role of ASCT in treatment of other solid tumors in children.
- In the section 3 - "Why only neuroblastoma? The original mistake" is about the trial which compared single transplant for consolidation vs none - the results of this trial should be more discussed (lines: 225-226)
- Are there currently studies with high-risk neuroblastoma patients treated without HDT and ASCT?
- What kind of treatment is used in reccurence neuroblastoma? Does the author know any trial for this group of patients? Do they include HDT and ASCT?
- The abbreviation MTD should be explain (line 119).
Author Response
Please see attached Word document

Round 2
Reviewer 2 Report
Dear Author
Thank you for your response.
I have the last question.
What kind of the review have you prepared - is it a narrative or scoping or systematic review?
It should be included in the article.
Author Response
Thanks for pointing this out. I stated the systematic review nature of the manuscript in the abstract.
